# Initial dose reduction of enzalutamide does not decrease the incidence of adverse events in castration-resistant prostate cancer

**Shunsuke Tsuzuki**[1], **Shotaro Nakanishi**[2], **Mitsuyoshi Tamaki**[2,3], **Takuma Oshiro**[2,3], **Jun Miki**[1], **Hiroki Yamada**[1], **Tatsuya Shimomura**[1], **Takahiro Kimura**[1]\*, **Nozomu Furuta**[1], **Seiichi Saito**[2], **Shin Egawa**[1]

1 Department of Urology, The Jikei University School of Medicine, Minato-ku, Tokyo, Japan, 2 Department of Urology, University of the Ryukyus, Graduate School of Medicine, Nishihara, Okinawa, Japan, 3 Department of Urology, Naha City Hospital, Naha, Okinawa, Japan

\* tkimura@jikei.ac.jp

**Data Availability Statement:** All relevant data are within the manuscript and its Supporting Information files.

## Abstract

### Background

There was no clear evidence whether the initial dose of enzalutamide affects the incidence of adverse events (AEs), and oncological outcome in patients with castration-resistant prostate cancer (CRPC).

### Methods

The clinical charts of 233 patients with CRPC treated with enzalutamide were reviewed retrospectively. After 1:3 propensity score matching (PSM), 124 patients were divided into a reduced dose group and a standard dose group, and the prostate specific antigen (PSA) response and the incidence of AEs were compared.

### Results

190 patients with CRPC initiated with standard dose enzalutamide were younger and better performance status compared with 43 patients beginning with reduced dose. After PSM, the baseline characteristics were not different between the standard and the reduced dose group. In the PSM cohort, the PSA response rate was significantly lower in the reduced dose group than in the standard dose group (-66.3% and -87.4%, p = 0.02). The incidence rates of AEs were not statistically different between the groups (22.6% and 34.4%, respectively, p = 0.24).

### Conclusion

Initiating treatment with a reduced dose of enzalutamide did not significantly decrease the incidence rate of AEs, and it showed poorer PSA response rate. There is no clear rationale for treating with a reduced initial dose of enzalutamide to reduce the incidence of AEs.

**Funding:** The authors received no specific funding for this work.

**Competing interests:** Shin Egawa is a paid consultant/advisor of Takeda, Astellas, AstraZeneca, Sanofi, Janssen, and Pfizer. Takahiro Kimura is a paid consultant/advisor of Astellas, Bayer, Janssen and Sanofi. This does not alter our adherence to PLOS ONE policies on sharing data and materials. Shunsuke Tsuzuki, Shotaro Nakanishi, Mitsuyoshi Tamaki, Takuma Oshiro, Jun Miki, Hiroki Yamada, Tatsuya Shimomura, Nozomu Furuta and Seiichi Saito declare no conflicts of interest. This does not alter our adherence to PLOS ONE policies on sharing data and materials.

## Introduction

Prostate cancer (PC) is the most common type of cancer among elderly men worldwide [1]. Overall, 7%–15% of patients with PC are initially diagnosed with metastatic PC and treated initially with androgen deprivation therapy (ADT) [2]. In addition, 20%–50% of non-metastatic PC patients treated with surgery or radiation have biochemical recurrence and are administered ADT [3]. Ultimately, these patients administered ADT develop castration-resistant prostate cancer (CRPC) within two years.

Enzalutamide, a next-generation androgen receptor (AR)-targeted agent, was approved for metastatic CRPC based on the results of phase 3 double-blind, randomised trials (PREVAIL and AFFIRM) [4, 5]. In these trials, enzalutamide significantly prolonged patient survival compared to placebo in both pre- and post-chemotherapy settings. More recently, phase 3 trials showed the efficacy of enzalutamide for non-metastatic CRPC and metastatic castration-sensitive PC [6–8]. A wide variety of adverse events (AEs), such as fatigue, decreased appetite, and hypertension, were reported in the trials. In both trials for metastatic CRPC, >90% of patients in the enzalutamide arm experienced AEs of any grade, and >40% experienced AEs of grade ≥3 [4, 5]. A similar tendency was observed in the subgroup analysis of a Japanese cohort in the PREVAIL trial. In detail, approximately 95% of patients in the enzalutamide arm had varying severity of AEs such as decreased appetite, weight loss, and fatigue [9]. Among such AEs, fatigue is the most frequent, with an incidence of 35.6% and 21.4% in the overall population and the Japanese subgroup of the PREVAIL trial, respectively [4, 9].

Patients sometimes require dose reduction or discontinuation of enzalutamide because of severe AEs [10]. Thus, the initial starting dose is sometimes reduced for patients with older age or poorer performance status (PS) in order to minimise the occurrence of AEs and to prolong the treatment period in real-world practice. A retrospective study of Japanese patients with CRPC reported that age >75 years was positively associated, and a lower initial dose of enzalutamide was negatively associated, with the occurrence of AEs [11]. Another retrospective study reported that efficacy of low-dose enzalutamide in terms of prostate specific antigen (PSA) response and PSA progression-free survival (PFS) was not statistically inferior to that of the standard dose in patients with metastatic CRPC aged ≥75 years [12]. However, significant biases should exist regarding patient background between the patients with the low and standard initial doses in both retrospective studies, and the influence of initial dose reduction on efficacy and safety remain to be well elucidated. Therefore, we aimed to investigate whether the initial dose of enzalutamide affects the incidence of AEs and oncological outcome in patients with CRPC by propensity score matching (PSM).

## Materials and methods

This study was approved by the Ethics Committee of both The Jikei University School of Medicine ((30-390(9411))) and University of the Ryukyus, Graduate School of Medicine. The requirement for informed patient consent was waived due to the study's retrospective design.

### Patients and study design

We retrospectively reviewed the clinical records of patients with CRPC treated with enzalutamide from June 2014 to December 2018 at The Jikei University Hospitals and University of the Ryukyus, Graduate School of Medicine and its affiliated institution. Patients were excluded from this study based on the following criteria: 1) lack of clinical information, 2) short follow-up periods (<1 month), and 3) prior treatment with abiraterone acetate. Ultimately, 233 patients were enrolled in this study. The initial dose of enzalutamide was decided based on an

individual patient's condition and the institutional policies, i.e., one institution (JU) used the standard dose and the other (RU) used a reduced dose as the starting dose in general. Patients were classified based on initial use of the standard dose or a reduced dose of enzalutamide, and the efficacy and safety of each dose were compared. The treatment sequence for CRPC was essentially decided according to the guidelines [13]. CRPC was defined according to the guidelines of the Prostate Cancer Clinical Trials Working Group 2 (PCWG2) [14]. PSA progression after enzalutamide treatment was defined according to the PCWG2 criteria [14]. In this study, the discontinuation of enzalutamide treatment either temporarily or permanently due to the occurrence of any AEs was used as the categorical variable in the Cox regression analyses.

## Statistical analysis

The patients' characteristics in the standard and reduced dose groups shown in Tables 1 and 2 were compared using the chi-square test and t-test. The propensity score for selecting the standard or reduced dose of enzalutamide was calculated; thereafter, 1:3 ratios of k-neighbour PSM was performed to reduce the variance of the baseline characteristics between the groups. The association of the initial dose of enzalutamide with PSA-PFS was estimated using the Kaplan-Meier method and the log-rank test. Univariate and multivariate Cox regression analyses were conducted to identify the independent factors for PSA-PFS. All data were analyzed using STATA 14 (Stata Corp., College Station, TX). Differences were considered significant if the two-sided p-values were <0.05.

## Results

The baseline characteristics of the 233 patients are listed in Table 1.

The patients' median age was 77 years (interquartile range [IQR], 70–81.5 years), and median PSA level before enzalutamide treatment was 14.4 ng/mL (IQR, 6.20–52.4 ng/mL). Among the 233 patients, 42% had undergone local therapies, such as prostatectomy or radiotherapy. Approximately 70% of the patients were initially treated with ADT plus bicalutamide, and the median time to CRPC was 18.5 months. Regarding the allocation of patients based on the initial dose of enzalutamide, 190 (81.6%), 25 (10.7%), 15 (6.4%), and 3 (1.3%) patients were administered 160, 120, 80, and 40 mg of enzalutamide, respectively. When stratifying the patients to the standard or reduced dose group, the patients in the reduced dose group were significantly older and had poorer PS than those in the standard dose group (Table 1). Although it was not statistically significant difference, the median time to CRPC in reduced dose group was shorter than that of the standard dose group (16.5 vs 19.5months, respectively).

The patients' baseline characteristics after PSM are shown in Table 2.

After 1:3 PSM, 31 and 93 patients were stratified to the reduced and the standard dose groups, respectively. The baseline characteristics were not significantly different between the groups.

In the PSM cohort, the PSA response after enzalutamide treatment is shown in Table 3 and Fig 1.

The PSA response rate after enzalutamide treatment was significantly higher in the standard dose group than in the reduced dose group (-87.4% and -66.3%, respectively, p = 0.02). The proportion of patients with a PSA decline of >90% was significantly higher in the standard dose group than in the reduced dose group (46.2% and 25.8%, respectively, p = 0.03). The median follow-up period was 17 months. Overall, 84 patients (67.7%) had PSA progression and 24 patients (19.4%) died due to any cause. The Kaplan-Meier curves for PSA-PFS according to the initial dose of enzalutamide are presented in Fig 2. The median time to PSA progression in the standard group tended to be longer than in the reduced dose group, but the

**Table 1. Baseline characteristics of 233 CRPC patients based on initial dose of enzalutamide.**

| Variables | Total Number (%) | Initial dose of Enzalutamide | | P value |
|---|---|---|---|---|
| | | 40-120mg | 160mg | |
| | | Number (%) | Number (%) | |
| Number of patients | 233 | 43 (18.5) | 190 (81.5) | |
| Median age (IQR) | 77 (70–81.5) | 81 (76–84) | 76 (69–80) | <0.01 |
| Median PSA(IQR) | 14.4(6.2–52.4) | 13.2(7.2–49.7) | 14.9 (5.2–54.6) | 0.55 |
| GS | | | | 0.73 |
| 6–7 | 40 (17.2) | 6 (14.0) | 34 (17.9) | |
| 8 | 38 (16.3) | 11 (25.6) | 27 (14.2) | |
| 9–10 | 107 (45.9) | 20 (46.4) | 87 (45.8) | |
| NA | 48 (20.6) | 6 (14.0) | 42 (22.1) | |
| PS | | | | <0.01 |
| 0–1 | 218 (93.6) | 34 (79.1) | 184 (96.8) | |
| 2–4 | 15 (6.4) | 9 (20.9) | 6 (3.2) | |
| cT stage | | | | 0.92 |
| Tx | 33 (16.3) | 6 (14.0) | 27 (14.2) | |
| T1-2 | 108 (44.2) | 19 (44.1) | 89 (46.9) | |
| T3-4 | 92 (39.5) | 18 (41.9) | 74 (38.9) | |
| cN stage | | | | 0.94 |
| 0 | 158(67.8) | 29 (67.4) | 129 (67.9) | |
| 1 | 75 (32.2) | 14 (32.6) | 61 (32.1) | |
| cM stage | | | | 0.66 |
| 0-1a | 95 (40.8) | 19 (44.2) | 76 (40.0) | |
| 1b-c | 138 (59.2) | 24 (55.8) | 114 (60.0) | |
| Median time to CRPC (IQR) | 18.5 (10–37) | 16.5 (9.5–46.5) | 19.5(10–36) | 0.59 |
| Prior DTX | 60 (25.8) | 7 (16.3) | 53 (27.9) | 0.09 |
| With Steroid Therapy | 43 (18.5) | 10 (23.3) | 33 (17.4) | 0.38 |
| With BMA | 45 (23.6) | 12 (27.9) | 33 (17.4) | 0.12 |

IQR = interquartile range, NA = not available, CRPC = castration resistant prostate cancer, DTX: docetaxel, BMA = bone modifying agent.

difference was not statistically significant (8 and 6 months, respectively, p = 0.10, Table 3). Univariate and multivariate Cox regression analyses were conducted to investigate the independent prognostic factors for PSA-PFS. The results are summarised in Table 4.

The initial dose of enzalutamide was not a significant predictive factor for PSA-PFS. In the multivariate analyses, time to CRPC within 12 months was the independent predictive factor for PSA progression (hazard ratio, 2.55; 95% confidence interval, 1.53–4.23; p<0.01).

During the follow-up period, the proportion of patients requiring a dose reduction due to AEs tended to be higher in the standard dose group than in the reduced dose group, even though the difference was not statistically significant (31.2% and 16.1%, respectively, p = 0.10). Moreover, the discontinuation rate of enzalutamide due to AEs was not significantly different between the groups (9.7% and 3.2%, respectively, p = 0.25). The AE profile of the patients is presented in Tables 5 and 6.

The incidence rates of AEs of any grade and grades 3–5 were 30.2% and 8.9%, respectively, in the entire cohort. Fatigue and appetite loss were frequent AEs observed in this cohort (Table 6). The incidences of AEs of any grade and grade 3–5 were not significantly different between the standard and reduced dose groups (34.4% and 22.6%, and 9.7% and 6.5%, p = 0.24 and 0.61, respectively).

**Table 2. Baseline characteristics of 124 CRPC patients after PSM.**

| Variables | Total | Initial dose of Enzalutamide | | P value |
|---|---|---|---|---|
| | | 40-120mg | 160mg | |
| Number of patients | 124 | 31 | 93 | |
| Median age (IQR) | 78.5 (74–82.5) | 79 (74–82) | 78 (74–83) | 0.55 |
| Median PSA(IQR) | 14.7 (7.3–46.3) | 13.6 (9.1–46.5) | 14.9 (7.1–45.0) | 0.49 |
| GS | | | | 0.95 |
| 6–7 | 19 (15.3) | 4 (12.9) | 15 (16.1) | |
| 8 | 19 (15.3) | 6 (19.4) | 13 (14.0) | |
| 9–10 | 59 (47.6) | 16 (51.6) | 43 (46.2) | |
| NA | 27 (21.8) | 5 (16.1) | 22 (23.7) | |
| PS | | | | 0.50 |
| 0–1 | 117 (94.4) | 30 (96.8) | 87 (93.5) | |
| 2–4 | 7 (5.6) | 1 (3.2) | 6 (6.5) | |
| cT stage | | | | 0.63 |
| Tx | 16 (12.9) | 3 (9.7) | 13 (14.0) | |
| T1-2 | 71 (60.5) | 20 (64.5) | 51 (54.8) | |
| T3-4 | 33 (26.6) | 8 (25.8) | 29 (31.2) | |
| cN stage | | | | 0.58 |
| 0 | 85 (68.5) | 20 (64.5) | 65 (69.9) | |
| 1 | 39 (31.5) | 11 (35.5) | 28 (30.1) | |
| cM stage | | | | 0.34 |
| 0-1a | 51 (41.1) | 15 (48.4) | 36 (38.7) | |
| 1b-c | 73 (58.9) | 16 (51.6) | 57 (61.3) | |
| Median time to CRPC (IQR) | 19 (10–37) | 15.0 (9–39) | 20.5 (12–36.5) | 0.58 |
| Prior DTX | 27 (21.8) | 6 (19.4) | 21 (21.5) | 0.71 |
| With Steroid Therapy | 25 (20.2) | 6 (19.4) | 19 (20.4) | 0.90 |
| With BMA | 30(24.2) | 7 (22.6) | 23 (24.7) | 0.75 |

IQR = interquartile range, NA = not available, CRPC = castration resistant prostate cancer, DTX: Docetaxel, BMA = bone modifying agent.

## Discussion

In the present study, we assessed the influence of the initial dose of enzalutamide on the oncological outcome and incidence of AEs in patients with CRPC. The results of the PSM cohort indicated that the PSA response after enzalutamide treatment was lower in the reduced dose group than in the standard dose group even though the PSA-PFS was not statistically different. On the other hand, the incidence and severity of AEs were not significantly different between the groups.

A dose escalation study of enzalutamide (30–360 mg per day) was conducted in previous preliminary and phase 1–2 trials, which found a dose-dependent increase of enzalutamide in the plasma [15, 16]. In fact, the dose dependency of enzalutamide in the percentage change of PSA from baseline was observed in the phase 1–2 study [16]. The proportion of patients with a PSA decline of >50% was lower in the patients who received 60 mg per day of enzalutamide than in those who received higher doses (>150 mg per day) [16]. Likewise, a dose dependency of enzalutamide was observed for the incidence of AE in the phase 1–2 study. Patients administered ≤150 mg per day of enzalutamide did not complain of fatigue or require discontinuation of treatment due to AEs [16]. Thus, the standard dose of enzalutamide was set as 160 mg per day.

**Table 3. Efficacy of enzalutamide treatment based on initial dose of enzalutamide.**

| Variables | Total | Initial dose of Enzalutamide | | P value |
|---|---|---|---|---|
| | | 40-120mg | 160mg | |
| Dose reduction | | | | 0.10 |
| None | 90 (72.6) | 26 (83.9) | 64 (68.8) | |
| Yes | 34 (27.4) | 5 (16.1) | 29 (31.2) | |
| Discontinued *1 | | | | 0.25 |
| None | 114 (81.9) | 30 (96.8) | 84 (90.3) | |
| Yes | 10 (8.1) | 1 (3.2) | 9 (9.7) | |
| % PSA decline (IQR) *2 | 83.3(34.7–97.4) | 66.3(24.1–94.9) | 87.4(46.2–97.7) | **0.02** |
| 50% PSA response *2 | 76 (61.3) | 16 (51.6) | 60 (64.5) | 0.20 |
| 90% PSA response *2 | 51 (41.1) | 8 (25.8) | 43 (46.2) | **0.03** |
| Median time to PSA nadir(IQR) | 2 (1–5) | 2(1–5) | 3(1–5) | 0.10 |
| Median time to PSA progression (range) | 6 (1–58) | 5 (1–24) | 8 (1–58) | 0.10 |

PSA = prostate specific antigen, IQR = interquartile range,

*1 due to any adverse events,

*2 at the best response.

However, there are a number of patients with severe AEs after administration of the standard dose of enzalutamide in real-world practice. Actually, 8.9% of patients had grade 3–5 AEs in the present study, even though the incidence was not higher than that of the previous phase 3 study [9]. To minimise the incidence and severity of AEs and to prolong the treatment period, some clinicians introduce enzalutamide with a reduced starting dose, even though the reduced dose might lead to insufficient treatment efficacy. The results of this study indicated that reducing the initial dose of enzalutamide did not decrease the incidence of AEs and might impair the efficacy.

Terada et al. retrospectively investigated 345 Japanese patients with CRPC and indicated that the lower initial dose of enzalutamide was negatively associated with the occurrence of AEs in the multivariate analysis [11]. However, the initial dose was decided according to the individual patient's condition, and significant biases should exist in this study. Regarding that point, the strength of the present study was that PSM was conducted to minimise the biases related to patients' backgrounds between the standard and reduced dose groups. The incidence of grade 3–5 AEs was generally very low in this study (9.7% and 6.5% in the standard and reduced dose groups, respectively). Nevertheless, 8.1% of the patients preferred to discontinue the enzalutamide treatment due to AEs. Regarding the median time to discontinuation of the enzalutamide treatment, no statistical difference was found between the groups (7.0 and 6.5

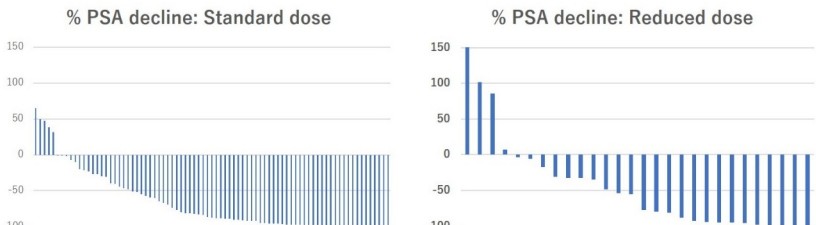

**Fig 1. Waterfall plots of PSA response after enzalutamide according to initial dose of enzalutamide in 124 CRPC patients.**

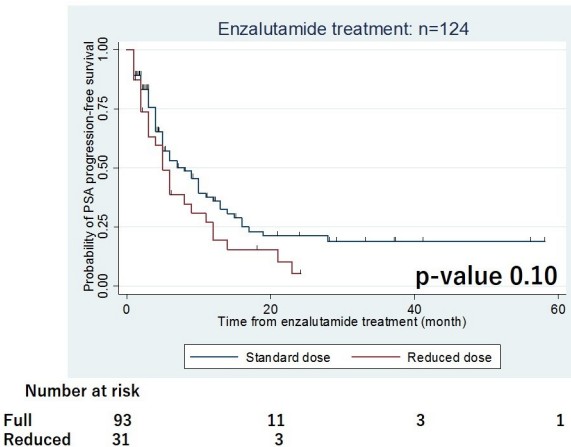

**Fig 2. Kaplan-Meier curves for PSA progression-free survival according to initial dose of enzalutamide in 124 CRPC patients.**

months, respectively; p = 0.48), suggesting that the initial dose reduction of enzalutamide might not prolong its treatment duration.

In this study, the PSA response rate after enzalutamide treatment was significantly higher, and the proportion of patients with a PSA decline of >90% was significantly higher, in the standard dose group than in the reduced dose group. The PSA response after enzalutamide was reported to be a predictor for radiographic PFS and OS [17]. The efficacy of enzalutamide might be impaired by reducing the initial dose, even though the PSA-PFS was not statistically different in this study, possibly because of the limited number of patients and events. Vinh-Hung et al. reported a retrospective study investigating the efficacy of low-dose enzalutamide in patients aged ≥75 years with metastatic CRPC and concluded that PSA response and PSA-PFS were not significantly different between the low dose and the standard dose groups

**Table 4. Univariable and multivariable Cox regression analyses for the prediction of PSA progression-free survival in 124 CRPC patients treated with enzalutamide.**

| Variable | PSA progression | | | |
|---|---|---|---|---|
| | Univariable | | Multivariable | |
| | HR (95%CI) | P-value | HR (95%CI) | P-value |
| Prior local therapy | 0.91(0.71–1.17) | 0.46 | | |
| Time to CRPC <12m | 3.04(1.89–4.91) | <0.01 | 2.55(1.53–4.23) | <0.01 |
| Age (continuous) | 1.00(0.97–1.04) | 0.80 | | |
| PSA[#1] (continuous) | 1.00(1.00–1.00) | <0.01 | 1.00(0.99–1.00) | 0.42 |
| PS2-4 (ref:0–1) | 0.74(0.18–3.02) | 0.68 | | |
| cT3-4 (ref:cTx-2) | 1.23(0.87–1.73) | 0.24 | | |
| cN1 (ref:N0) | 1.92(1.23–3.00) | <0.01 | 1.55(0.96–2.51) | 0.08 |
| cM1b-c (ref: M0-1a) | 1.15(0.75–1.77) | 0.53 | | |
| Prior DTX (ref:none) | 2.16(1.31–3.54) | <0.01 | 1.58(0.86–2.88) | 0.14 |
| Steroid (ref:none) | 1.67(1.00–2.77) | 0.05 | 1.28(0.72–2.26)- | 0.41- |
| BMA (ref:none) | 1.18(0.73–1.93) | 0.49 | | |
| Initial Enz dose (ref:standard) | 1.45(0.96–2.31) | 0.12 | - | - |

HR = Hazard Ratio; CI = Confidence Interval; #1: PSA before enzalutamide treatment, BMA = bone modifying agent, Enz: enzalutamide, DTX: docetaxel.

**Table 5. Any grade of adverse events (AE) based on initial dose of enzalutamide.**

| Variables | Total | Initial dose of Enzalutamide | | P value |
|---|---|---|---|---|
| | | 40-120mg | 160mg | |
| Number of patients | 124 | 31 | 93 | |
| Incidence of AE | | | | |
| All grade | 39 (31.5 | 7 (22.6) | 32(34.4) | 0.24 |
| Grade3-5 | 11 (8.9) | 2 (6.5) | 9 (9.7) | 0.61 |

[12]. However, this retrospective study was limited in that 55 patients were stratified to 16 low dose and 43 standard dose groups. Thus, we considered that introducing the enzalutamide with a reduced dose might impair the efficacy. Dose reduction or temporary discontinuation of enzalutamide should be considered if the patients develop AEs.

In this study, the multivariate analysis revealed that the time to CRPC within 12 months was the independent predictive factor for poorer PSA-PFS. This finding was consistent with those of previous reports [5, 18, 19].

This study has several limitations. First, it was a retrospective study with a small sample size; thus, selection bias might have been introduced even though PSM was conducted to minimise the bias. Second, AEs were observed in only 31.5% of the patients, which might be due to the retrospective design of this study. However, the data for patients with AEs of grade 3 or higher should be reliable. Third, the short follow-up period might have affected the results of the oncological outcome. Finally, due to the small number of patients analyzed, we did not divide the patients with CRPC according to the history of prior chemotherapy. Despite these limitations, the present work is a unique study that investigated the relation of the initial dose of enzalutamide to the oncological outcomes and incidence of AEs in the PSM cohort. We believe that our data will be useful in daily clinical practice. A further external validation cohort and prospective study are warranted in the future.

## Conclusions

Initiating treatment with a reduced dose of enzalutamide did not significantly decrease the incidence rate of AEs, and it showed poorer PSA response rate. There is no clear rationale for treatment with a reduced initial dose of enzalutamide to reduce the incidence of AEs.

**Table 6. Profile of adverse events (AE) based on initial dose of enzalutamide.**

| Initial dose of enzalutamide | 40-120mg | | 160mg | |
|---|---|---|---|---|
| Adverse Event | All grade | G3-5 | All grade | G3-5 |
| Fatigue | 2 (6.5) | 2 (6.5) | 15(16.1) | 4(4.3) |
| Appetite loss | 4 (12.9) | 0 | 10 (10.8) | 3 (3.2) |
| Rush | 0 | 0 | 1 (1.1) | 0 |
| AST/ALT increased | 1 (3.2) | 0 | 1 (1.1) | 0 |
| Nausea | 0 | 0 | 5 (5.4) | 0 |
| Diarrhoea | 0 | 0 | 3 (3.2) | 0 |
| Hypertension | 0 | 0 | 1 (1.1) | 1(1.1) |
| Dysgeusia | 0 | 0 | 1(1.1) | 0 |
| Edema | 0 | 0 | 2 (2.2) | 0 |
| Stiffness | 0 | 0 | 1(1.1) | 1(1.1) |

## Supporting information

**S1 File. Anonymized data set.**
(XLSX)

## Author Contributions

**Conceptualization:** Shunsuke Tsuzuki, Shotaro Nakanishi, Tatsuya Shimomura, Takahiro Kimura, Seiichi Saito, Shin Egawa.

**Data curation:** Shunsuke Tsuzuki, Shotaro Nakanishi, Mitsuyoshi Tamaki, Takuma Oshiro, Jun Miki, Hiroki Yamada, Tatsuya Shimomura, Takahiro Kimura, Nozomu Furuta.

**Formal analysis:** Shunsuke Tsuzuki.

**Investigation:** Shunsuke Tsuzuki, Takahiro Kimura, Seiichi Saito, Shin Egawa.

**Project administration:** Shunsuke Tsuzuki, Shotaro Nakanishi, Takahiro Kimura, Seiichi Saito, Shin Egawa.

**Supervision:** Takahiro Kimura, Seiichi Saito, Shin Egawa.

**Validation:** Shunsuke Tsuzuki.

**Writing – original draft:** Shunsuke Tsuzuki.

**Writing – review & editing:** Shunsuke Tsuzuki, Shotaro Nakanishi, Mitsuyoshi Tamaki, Takuma Oshiro, Jun Miki, Hiroki Yamada, Tatsuya Shimomura, Takahiro Kimura, Nozomu Furuta, Seiichi Saito, Shin Egawa.

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
