## [Decision Letter · Decision Letter 0]

20 Aug 2021

PONE-D-21-17359

Initial dose reduction of enzalutamide does not decrease the incidence of adverse events in castration-resistant prostate cancer

PLOS ONE

Dear Dr. Tsuzuki,

Thank you for submitting your manuscript to PLOS ONE. After careful consideration, we feel that it has merit but does not fully meet PLOS ONE’s publication criteria as it currently stands. Therefore, we invite you to submit a revised version of the manuscript that addresses the points raised during the review process.

We look forward to receiving your revised manuscript.

Kind regards,

Henry Woo

Academic Editor

PLOS ONE

Journal Requirements:

I have read the journal's policy and the authors of this manuscript have the following competing interests:Shin Egawa is a paid consultant/advisor of Takeda, Astellas, AstraZeneca, Sanofi, Janssen, and Pfizer. Takahiro Kimura is a paid consultant/advisor of Astellas, Bayer, Janssen and Sanofi. 

4. We note you have included a table to which you do not refer in the text of your manuscript. Please ensure that you refer to Table 6 in your text; if accepted, production will need this reference to link the reader to the Table.

Reviewers' comments:

Reviewer's Responses to Questions

**Comments to the Author**

1. Is the manuscript technically sound, and do the data support the conclusions?

Reviewer #1: Yes

2. Has the statistical analysis been performed appropriately and rigorously? 

Reviewer #1: Yes

3. Have the authors made all data underlying the findings in their manuscript fully available?

Reviewer #1: Yes

4. Is the manuscript presented in an intelligible fashion and written in standard English?

Reviewer #1: Yes

5. Review Comments to the Author

Reviewer #1: The authors have explored the impact of the initial dose of enzalutamide on the treatment response and incidence of AEs in patients with CRPC. They report that initial dose modification of enzalutamide did not decrease the incidence of AEs and might impair the efficacy. The topic is clinically significant as the patients are getting older, as written in the manuscript. The manuscript is concise and well-written. Some points to be clarified are as below.

1) In the univariate and multivariate analysis, 'Time to CRPC <12m' is independent risk factor for the prediction of PSA progression-free survival (Table 4). Then, I suppose some patients who had long, durable response with previous ADT (Time to CRPC >12m) might be treated with initial dose reduction. In contrast, some with resistant to ADT might be treated without dose modification. So , Could you check and provide subgroup analysis between these two group according to Time to CRPC?

2) It would be good to provide relative dose intensity if you can calculate from the subjects.

3) In Table 2, the Median time to CRPC (IQR) of the two groups seems different (median time 10.5 vs. 20.5). Although, P-value is not statistically significant. I recommend checking this value and provide other values, such as mean time or distribution if possible.

4) Please add abbreviations to each table.

6. PLOS authors have the option to publish the peer review history of their article (what does this mean?). If published, this will include your full peer review and any attached files.

Reviewer #1: No

---

## [Author Response · Author response to Decision Letter 0]

6 Sep 2021

Thank you for reviewing our manuscript and offering us the opportunity to submit the revised one. We would also like to thank PLOS ONE Editorial Board that provides an opportunity to improve the manuscript with their helpful comments. We have revised the manuscript according to the reviewers’ suggestions.

---

## [Editor Report · Decision Letter 1]

20 Sep 2021

Initial dose reduction of enzalutamide does not decrease the incidence of adverse events in castration-resistant prostate cancer

PONE-D-21-17359R1

Dear Dr. Tsuzuki,

We’re pleased to inform you that your manuscript has been judged scientifically suitable for publication and will be formally accepted for publication once it meets all outstanding technical requirements.

Kind regards,

Henry Woo

Academic Editor

PLOS ONE

---

## [Editor Report · Acceptance letter]

23 Sep 2021

PONE-D-21-17359R1 

Initial dose reduction of enzalutamide does not decrease the incidence of adverse events in castration-resistant prostate cancer 

Dear Dr. Tsuzuki:

I'm pleased to inform you that your manuscript has been deemed suitable for publication in PLOS ONE. Congratulations! Your manuscript is now with our production department. 

Kind regards, 

on behalf of

Prof. Henry Woo 

Academic Editor

PLOS ONE